# The Feasibility of Early Alzheimer’s Disease Diagnosis Using a Neural Network Hybrid Platform

**DOI:** 10.3390/bios12090753

**Published:** 2022-09-13

**Authors:** Xinke Yu, Siddharth Srivastava, Shan Huang, Eric Y. Hayden, David B. Teplow, Ya-Hong Xie

**Affiliations:** 1Department of Materials Science and Engineering, University of California, Los Angeles, CA 90095, USA; 2Department of Neurology, David Geffen School of Medicine at UCLA, University of California, Los Angeles, CA 90095, USA; 3Jonsson Comprehensive Cancer Center, University of California, Los Angeles, CA 90095, USA

**Keywords:** SERS, Alzheimer’s disease, biosensing, Raman spectroscopy, machine learning, neural networks, disease diagnosis, materials science, nanomaterials

## Abstract

Early diagnosis of Alzheimer’s Disease (AD) is critical for disease prevention and cure. However, currently, techniques with the required high sensitivity and specificity are lacking. Recently, with the advances and increased accessibility of data analysis tools, such as machine learning, research efforts have increasingly focused on using these computational methods to solve this challenge. Here, we demonstrate a convolutional neural network (CNN)-based AD diagnosis approach using the surface-enhanced Raman spectroscopy (SERS) fingerprints of human cerebrospinal fluid (CSF). SERS and CNN were combined for biomarker detection to analyze disease-associated biochemical changes in the CSF. We achieved very high reproducibility in double-blind experiments for testing the feasibility of our system on human samples. We achieved an overall accuracy of 92% (100% for normal individuals and 88.9% for AD individuals) based on the clinical diagnosis. Further, we observed an excellent correlation coefficient between our test score and the Clinical Dementia Rating (CDR) score. Our findings offer a substantial indication of the feasibility of detecting AD biomarkers using the innovative combination of SERS and machine learning. We are hoping that this will serve as an incentive for future research in the field.

## 1. Introduction

Alzheimer’s disease (AD) affects millions of people worldwide and is increasing drastically due to the aging population [1,2,3,4]. The disease affects older adults and is the most common cause of dementia [5]. No cure or disease-modifying therapy exists [6,7,8]. The prerequisite for finding a cure is to have an objective diagnosis platform or a biomarker that can diagnose the disease. No consensus on a platform or biomarker currently exists. Instead, diagnoses rely on preanalytical, analytical, and postanalytical variables which affect CSF biomarker concentrations [9,10,11,12,13,14]. Pre-analytical factors could be clinical variables such as age, medical history, apolipoprotein E (APOE) genotype, and operator-influenced variables (handling and storage procedures, CSF sampling material), and these are responsible for 50% of total variability [12,15]. Analytical factors are related to the assay, such as the operating procedures, assay manufacturing procedures, and the technician’s skill and knowledge. Furthermore, cognitive testing and a variety of surrogate biomarkers, including brain imaging, proteins in the cerebrospinal fluid (CSF), proteins in the blood, and genetic profiling [7,16,17] add more layers of complications. As a result, diagnosing AD patients is a lengthy and costly process, which impedes patient care [8]. The development of biomarkers that allow the detection of AD during the pre-symptomatic phase is critical to the discovery and development of effective AD diagnoses and treatments. Efforts are underway to discover unknown biomarkers and to develop biomarkers comprising combinations of proteins [18,19,20,21].

CSF provides a means to assess neurodegenerative processes in the brain through the identification of disease-associated molecules. These molecules most often are proteins or protein fragments, including the amyloid β-protein (Aβ) and tau [20,22,23,24,25,26,27], which are amenable to analysis using a variety of techniques, including surface-enhanced Raman spectroscopy (SERS) [28,29]. The raw spectral information collected from CSF covers a huge multi-dimensional dataset. To analyze such sets, machine learning methods are increasingly being used because of their low cost, speed, and sensitivity. Convolutional neural networks (CNN) are one such approach and have been shown to be successful in multiple fields, including disease diagnosis [30,31,32]. Recently, several studies on AD diagnosis have been conducted using deep learning techniques, and these studies have mainly focused on brain imaging differentiation, which can only take place after the onset of diseases [32,33,34,35,36]. Here, we propose a novel method for biomarker detection that combines SERS with CNN to analyze disease-associated changes in the CSF. This method yields “fingerprints” of samples, allowing ready discrimination of CSF samples among normal individuals and those with neurological disorders.

## 2. Materials and Methods

### 2.1. Substrate Preparation

Fabrication of the platform was based on sphere lithography [26]. A periodic gold nano pyramid structure, with a diameter of 200 nm, was fabricated by a wafer-scale, bottom-up templating technology [37]. Close-packed monolayer polystyrene balls with a diameter of 200 nm, spin-coated on (001) silicon wafers, served as templates. Monolayer graphene was grown by chemical vapor deposition (CVD) and the solution was transferred onto the gold-tipped surface using a polymethyl methacrylate (PMMA) backing, followed by PMMA removal subsequent to the transfer. Such platforms can be fabricated with user-determined areas. We typically used platforms of ~1 cm^2^. The pyramids formed a quasi-periodic array with a hexagonal arrangement that was uniformly distributed across the entire sample surface area of 1 cm × 1 cm. Due to the manner in which the pattern was generated (self-assembly of polystyrene balls), variations in the spacing between pyramids and the sizes of the pyramids themselves can appear. This variance was estimated to be ±30 nm.

### 2.2. Sample Preparation

CSF samples were collected and diluted by a factor of 100 and applied to the SERS platform.

### 2.3. Raman Spectroscopy

Spectra were acquired using a Renishaw inVia microscope under ambient conditions. The excitation wavelength was 785 nm and the He-Ne laser power was 0.5 mW. The grating used was 1800 lines/mm, and the objective lens used was 50×. We scanned the entire region on the platform occupied by the samples using Raman mapping with a step size of 3 μm (i.e., independent areas of 9 µm^2^ each), as shown in Figure 1. Raman data were analyzed using Renishaw’s WiRE 4.2 software (Gloucestershire, UK), which automatically subtracts the baseline signal and removes noise. A minimum of 80 spectra were acquired and averaged for each sample.

### 2.4. Patient Information

Thirty CSF samples were obtained from the University of California, Irvine, Institute for Memory Impairment and Neurological Disorders, Alzheimer’s Disease Research Center (UCI MIND, ADRC). Several characteristics of the patients’ CSF had already been measured, using standard procedures among the ADRCs. These included the levels of Aβ42, total tau, and phospho-tau, as well as the mini-mental state exam (MMSE) and the clinical dementia rating (CDR). A summary of the patient data is shown in Table 1.

### 2.5. Hierarchical Clustering Algorithm

The analysis in this work was carried out using R. R was chosen for the Hierarchical Clustering Algorithm (HCA) implementation because it is suitable for statistical learning, and having powerful libraries for data experiment and exploration. Several functions are available in R for HCA, and “hclust” was used because it fulfilled our need for agglomerative hierarchical clustering. The distance matrix was calculated by the function “dist”. The linkage method was changed by adding “method” in the “hclust” function. The results were plotted in dendrogram format with complete linkage. The spectra analyzed using HCA were the average spectra of each CSF sample.

### 2.6. Convolutional Neural Network

A one-dimensional CNN was used to process and classify the SERS spectral data. CNN was used instead of other popular models, such as support vector machines because other works in the field have highlighted that CNN’s offer a much better alternative pipeline [38]. To elaborate, a standard Raman data analysis pipeline includes steps for cosmic ray removal, smoothing, and baseline correction, and each part has to be trained by manual crafting or independently. However, CNN offered a much more simplified pipeline with higher model accuracy [38]. The convolutional layers of our model used the ReLU nonlinear activation function, and the convolutional layers were connected by max-pooling layers which down-sampled the feature maps. The output of the last max-pooling layer was fed to two consecutive fully connected layers to give the final classification result [38]. During the training process, the scalar sum of weighted losses was used to train the CNN model.

We started with a dataset size of over 1200 spectra, i.e., at least 80 spectra per sample. To increase the training set, some methods of augmenting the training data were applied: (i) random shifting of each spectrum by a few (1–2) wavenumbers; (ii) introduction of random noise into each spectrum; and (iii) randomly producing linear combinations of spectra collected from the same mapping procedure. The adaptive gradient algorithm (Adagrad) was used for gradient-based optimization while training the model, and early stopping was applied to prevent overfitting. In the model, the learning rate was adapted component-wise by incorporating knowledge of past observations. The Python version used was 3.6.8, and the neural network model was implemented in Tensorflow 1.11.0 (Mountain View, CA, USA), Google’s open-source software library for machine learning. The R version used was 3.6.1. We used the Anaconda coding environment for data analysis and used the matplotlib Python library for plotting.

## 3. Results and Discussion

### 3.1. Reproducibility Analysis

To determine the feasibility of the method, we used 5 distinct sets of CSF samples, where each set contained three replicates. Spectra were acquired from each of the three replicates of each sample set. The study was double-blinded. Neither those providing the samples nor the person carrying out the spectroscopy and data analysis knew the provenance of the samples.

SERS spectra were acquired in the wavenumber range of 550–1650 cm^–1^. SERS mappings with a step size of 3 μm, covering a 9 µm^2^ area, were carried out. A minimum of 80 spectra were acquired and averaged for each sample. All 15 average spectra are shown in Figure 2. Analysis of the spectra revealed that they could be clustered into five groups of three spectra each. This was carried out by using an unsupervised hierarchical clustering algorithm (HCA), and the details are presented in the subsequent paragraphs and Figure 3. The sample identifiers are shown in a table (Figure 2f). After the clustering had been carried out, the samples were unblinded. Disease information is shown in the rightmost column of the table. The method was reproducible and sensitive. Reproducibility was indicated by the clustering of replicates from the same sample. Sensitivity was indicated by the fact that all five CSF samples were clearly differentiated from each other. Of particular importance was the fact that even individuals with the same disease could be differentiated from one another.

Hierarchical cluster analysis (HCA) was used to determine the similarity and differences among the 15 samples quantitatively. Integration normalization was carried out when comparing the samples. The wavenumber of each of the SERS peaks represents one dimension and the peak height represents the corresponding value in that dimension. We grouped the samples using a single linkage of the aggregation HCA. The possible sources of the major peaks that contributed to the formation of different clusters are the—850 cm^−1^ peak from alanine and/or the 870 cm^–1^ peak from glutamic acid and serine, the 1000 cm^–1^ phenylalanine peak, the 1171 cm^–1^ tyrosine peak, the 1465 cm^−1^ lipid peak, and the 1555 cm^−1^ tryptophan [39,40,41].

HCA clustered the 15 samples into five clearly separated groups (Figure 3). The degree of similarity among the samples is indicated by the proximity to zero (horizontal axis) of the lines connecting them. The closer the connecting line is to 0, the more similarities the two samples share. This analysis, which revealed full concordance between sample provenance and the clustering, demonstrates the method’s accuracy and feasibility.

We next studied the ability of the method to distinguish between clinically evident AD and pre-clinical AD (FAD samples) using a “leave-one-out” validation procedure. We trained with all samples save one. We then analyzed the remaining sample. We repeated this procedure until every spectrum was left out once, then the average accuracy across all the data was computed. Leave-one out validation was used mainly because of its low evaluation error and the ease with which it can be used for small sample sizes. The results showed an accuracy of 97.7% for the spectrum of the FAD sample and 93.3% for the spectra of the AD sample, which suggests there was good uniformity over the SERS spectral data of these samples and that they were differentiable. The results of this double-blind study are a significant step forward in establishing the power of the platform for analyzing patient samples, despite intrinsic biological variability.

### 3.2. Disease Diagnosis

We also performed a leave-one-group-out evaluation on the dataset because a leave-one-out evaluation cannot determine whether our model is capable of exploring the relationship between the SERS testing result for a subject’s CSF sample and the subject’s diagnosed syndrome (normal or abnormal). This is because the spectral data collected from the same CSF sample are likely to be dependent on that individual sample, thus forming a group of dependent data, so that for an unseen sample that is yet to be classified, the correlation information of its spectral data should not be given to the model during training. Therefore, for each round of evaluation, we needed to make sure that all the spectral data in the test set came from groups that were not represented at all in the corresponding training set. Since we had a total of 17 CSF samples, a leave-one-group-out evaluation was a suitable approach for us to know whether our model could be generalized well to the unseen samples. During each evaluation, the entire SERS spectral data collected from one CSF sample were used for testing, while the rest of the data were used for training. This kind of evaluation was repeated until every group of spectral data had been left out once.

The final classification result also took the group dependency into consideration. In each round of evaluation, after the trained CNN made predictions on every testing spectrum, all the predictions were then combined through a majority vote to produce the final prediction, i.e., the class with the highest percentage of predictions was considered to be the predicted class of the testing sample. We used the percentage of predictions leading to the predicted class as a score to represent the likelihood that the testing sample belonged to the predicted class. An overall 94% diagnostic rate was achieved (Table 2). The normal sample had an accuracy of 100% (8/8) and an average score of 89.2, whereas the AD samples had an accuracy of 88.9% (8/9) and an average score of 72.0.

Within this table, we are able to see that Sample R was the only one with a prediction score less than 50, indicating we were not able to tell if R was a diseased sample. To further understand this sample, we referred to her cognitive test score. Her MMSE score was 25, which should be diagnosed as normal. However, her CDR test score was far away from normal (9 for CDRSUM and 1 for CDRGLOB) [14]. These scores present mixed information for diagnostics and inevitably influenced our test results.

To further understand more complicated situations, such as FAD-related patients, we performed the same test based on our training of normal and diseased patients. FAD-negative samples (pre-clinical FAD) had an accuracy of 4/4 (100%) and an average score of 91.25; FAD-positive samples had an accuracy of 4/5 (80%) and an average score of 80.0. The results are shown in Table 3.

Sample D had a test score of 49, indicating we are not able to accurately define whether she was diseased or not. When referring to her medical scores, we found that she had a MMSE score of 28 and CDRSUM of 0.5, indicating that she has mild dementia [14]. More analysis needs to be carried out to see the relationship between our test score and the symptoms or severity of the patients.

### 3.3. Correlation Analysis

To further understand the feasibility of our diagnostic results, we performed a correlation analysis for the diagnostic index and all the AD-related medical parameters. Several parameters were included in the analysis: sex, age, the levels of several biomarkers (Aβ42, t-tau, and p-tau), and several cognitive test scores (MMSE, CDRSUM, and CDRGLOB).

To better analyze the correlation, we first needed to deal with the missing data. In the clinical information provided, the t-tau information of Samples N, R, R, S, T, W, and X were missing, and the p-tau information of Sample R was missing. According to the literature, the t-tau and p-tau levels in CSF are highly correlated, and in the data provided, we saw a highly linear relationship between the concentration of the two types of tau protein with a correlation factor of r = 0.8883. The prediction model can be easily shown by regression:P = 0.1134T + 35.28,(1)
where P is the concentration for p-tau protein and T is the concentration for t-tau protein. We are able to fill in the missing t-tau levels with this correlation. For the missing data in Sample R, a detailed correlation analysis was carried out, showing that the parameter with the highest correlation with the tau protein level was MMSE, and the factor was only r = 0.47. As a result, we eliminate Sample R during the correlation analysis. The correlation coefficients between all the biomedical parameters are shown in Figure 4.

We can see from the figure that our prediction index was highly correlated to all three cognitive test scores. The correlation coefficients were r = 0.79 with MMSE, r = −0.88 with CDRSUM, and r = −0.87 with CDRGLOB. Considering that the CDR scores are taken as an accurate method of diagnosing AD, our prediction index is accurate as a diagnostic. We also observe from the figure that the correlation between single biomarkers (Aβ protein, tau protein) and the cognitive test score (MMSE and CDR) is relatively low. The highest correlation coefficient of a single biomarker was r = 0.47 (between t-tau and MMSE), which is still a low value. This highlights the fact single biomarkers cannot be used to accurately diagnose AD. However, our SERS/AI system has the ability to carry out combinatorial analyses, which allow for much more accurate disease diagnoses, as indicated by the high correlation with the CDR scores.

## 4. Conclusions

We have presented here a hybrid system of SERS and CNN for diagnosing Alzheimer’s disease with high accuracy. We have achieved excellent reproducibility in double-blind experiments and 92% accuracy in disease diagnosis. The correlation analysis demonstrated that our diagnostic system is more accurate than single-biomarker analyses. The SERS neural network is a novel platform that promises accurate, reliable, and rapid clinical diagnosis of AD. Even though the current study uses a small sample size (due to the limited availability of AD patient CSF samples), the very high accuracy obtained in the double-blind experiments is very encouraging. Our study, therefore, provides an indication of the feasibility of a biological test for AD diagnosis. We are hoping that this will serve as an incentive for much more extensive follow-up research by the Alzheimer’s research community at large. In the future, we expect this approach to allow early-stage AD diagnosis, and we hope this study will be a stepping stone for the SERS/AI technology to find use as a prognostic device and in clinical trials. To that end, subsequent studies will involve larger patient sample sizes, and demonstrate the sensitive and specific diagnostic capability using blood samples.

## Figures and Tables

**Figure 1 biosensors-12-00753-f001:**
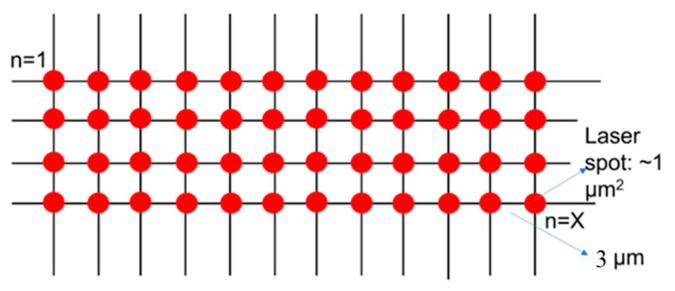
SERS mapping setup.

**Figure 2 biosensors-12-00753-f002:**
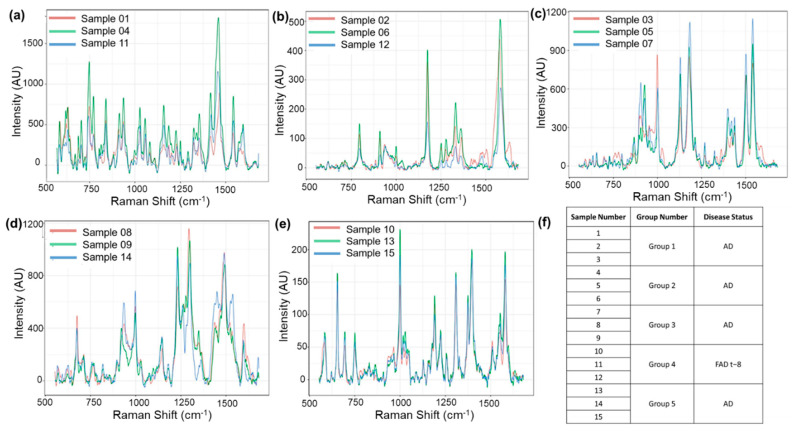
(**a**–**e**): SERS spectra of the 15 CSF samples grouped by spectral features. (**f**): Sample information for the 15 CSF samples. The average spectra were plotted in groups according to the HCA agglomerative clustering results.

**Figure 3 biosensors-12-00753-f003:**
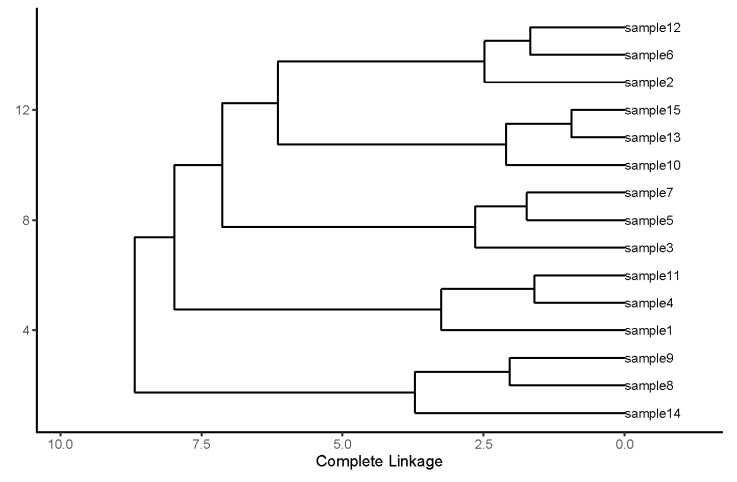
HCA analysis for the 15 unknown CSF samples, showing highly accurate clustering results.

**Figure 4 biosensors-12-00753-f004:**
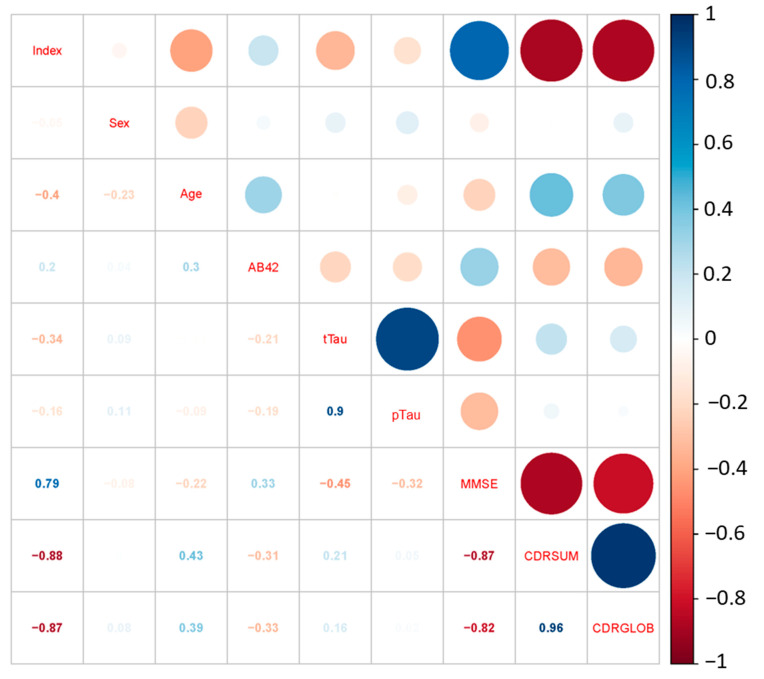
Correlation analysis between prediction score and bio-medical matrix.

**Table 1 biosensors-12-00753-t001:** Patient information summary.

	Healthy	Dementia	FAD+	FAD−
# of cases	10	9	5	4
Male/female	3/7	4/5	3/2	3/1
Age (years)	76.6 (+/−5.5)	79 (+/−4.9)	36 (+/−12.9)	34 (+/−14.8)
Adjusted age	NA	NA	−10 (+/−10.6)	NA
CSF Aβ42 (pg/mL)	645.6 (+/−353.0)	375.9 (+/−305.8)	186.2 (+/−60.4)	418.8 (+/−174.9)
CSF Total tau (pg/mL)	364.9 (+/−265.3)	570.6 (+/−529.4)	516.9 (+/−363.3)	312.1 (+/−266.8)
CSF phospho-tau (pg/mL)	83.8 (+/−43.7)	87.2 (+/−43.6)	99.2 (+/−50.8)	73.7 (+/−39.8)
MMSE (0–30)	29.9 (+/−0.3)	19.6 (+/−3.6)	25 (+/−7.9)	28.8 (+/−0.5)
CDR—sum of boxes (0–18)	0.1 (+/−0.2)	9.1 (+/−2.0)	1.6 (+/−3.0)	0.25 (+/−0.5)
CDR—global (0–3)	0.1 (+/−0.2)	1.44 (+/−0.5)	0.2 (+/−0.45)	0.13 (+/−0.25)

**Table 2 biosensors-12-00753-t002:** Test score for 17 non-FAD samples.

Sample	Label	Score
A	Normal	91.85
F	Normal	85.94
G	Normal	84.09
H	Normal	92.86
I	Normal	80.22
M	Normal	90.91
X	Normal	87.5
AA	Normal	100
B	Dementia	54.55
N	Dementia	88.64
O	Dementia	86.11
R	Dementia	31.25
S	Dementia	100
U	Dementia	89.80
W	Dementia	57.81
Y	Dementia	65.08
AB	Dementia	75

**Table 3 biosensors-12-00753-t003:** Test score for FAD-related samples.

Sample	Label	Score
D	FAD(+), −19	49
E	FAD(+), −5	100
K	FAD(−), −11	85
L	FAD(−), −17	100
P	FAD(−), 0	85
Q	FAD(+), 4	53
Z	FAD(+), −8	100
AC	FAD(+), −22	98
AD	FAD(−), −18	95

## Data Availability

Not applicable.

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
