# Peer review of "The Feasibility of Early Alzheimer’s Disease Diagnosis Using a Neural Network Hybrid Platform"

_biosensors, 2022, doi:10.3390/bios12090753_

Round 1

Reviewer 1 Report

In the submitted manuscript entitled “The Feasibility of Alzheimer's Disease Early Diagnosis Using a Neural Network Hybrid Platform”, the authors acquired 15 spectra of cerebrospinal fluid by surface enhanced raman spectroscopy and analyzed by a clustering method, showing agreement with diagnostic results. While early diagnosis of Alzheimer’s disease is of great importance, the presentation here is rather weak with a series of loopholes. Moreover, there are many typos across the manuscript, such as the “pTa” which should be “p-Tau”, in their main figure?! Therefore, I cannot recommend the publication of such a manuscript to Biosensors.

MAJOR

1. How did the authors determine the number of patients for statistical significance? 15 patients are far from making a solid conclusion or training a machine learning algorithm.

2. The authors have to mention clearly how the spectra were acquired.

3. the authors need to discuss the reproducibility issue of SERS technique, i.e. the same test subject would produce different spectra at different measurement trials. Such issue would obviously affect the accuracy of the diagnosis.

4. Machine learning has been a hot topic recently. But in this work, it is not really a machine learning approach. The authors need to discuss the training process of their CNN and list the details such as the fidelity score of each epoch. With so little data, it is hard to believe a machine learning network would work at all.

5. Another major concern is the variation of the spectra in figure 1. Why is there such huge difference in between groups, and so small difference within each group? These spectra just doesn’t make any sense.  Please attribute each of the peak to a real biomolecule.

Author Response

We thank the reviewer for their suggestions and comments. Please see the attachment.

Reviewer 2 Report

In this manuscript, the authors proposed a method for early diagnosis of Alzheimer's disease using a neural network hybrid platform. SERS and CNN are combined for biomarker detection to analyze disease-associated changes in CSF. The experimental results demonstrate the effectiveness of the method. This research has certain clinical application value. There are some issues that need to be addressed before it is ready to be published. The followings are my comments and suggestions.

1. In Section 2.5, why choose the function “hclust” for the HCA analysis?

2. How to select the CNN model in the method proposed in this paper should be explained in Section 2.6.

3. The quality of Figure 1 needs to be improved.

Author Response

(The authors gave the same response as above.)

Reviewer 3 Report

Dear Authors.

This paper is good paper (The Feasibility of Alzheimer's Disease Early Diagnosis Using a Neural Network Hybrid Platform). But, I decision reconsider after major revision.

Strength of this paper included:

The strength of the paper included: the topic is good and interesting.

Weakness of this paper:

#1. Introduction

The "Introduction" information presented is not new.

I recommend additional/rewrite "Introduction".

#2. Code and Simulation

What kind of Simulation/Coding language (version) did you use?

-Please add/more version and description.

-Simulation/Coding language environment.

-Environment explanation/presentation more.

#3. Abstract and contribution: Poor Abstract.

I recommend additional/rewrite "Abstract and contribution".

-Contribution need supported by data and result.

#4. Methods.

-They should be described with sufficient detail to allow others to replicate and build on published results. New methods and protocols should be described in detail while well-established methods can be briefly described and appropriately cited. Give the name and version of any software used and make clear whether computer code used is available. Include any pre-registration codes (Methods).

#5. Related work: Improve (Study more)

-Also, more to 10 new papers (Journal) published from 2019~2022 by major publishers such as Nature publishing group, Science: American Association for the Advancement of Science group, Springer, IEEE, ACM, Elsevier, and Wiley.

-More New Journal (2019~2022) papers.

#6. Results

Results need clearly.

-Need nice story of results.

#7. Other

Scientific Soundness: Low.

#8. English

-English language and style are fine/minor spell check required.

#9. Conclusion

-You need to write more of the conclusion part. 

-Future work" write more. (Must be improved) 

-Conclusions supported by the results. (Write more)

#10. Figure quality: Poor

Also, I recommend drawing a new Figure.

Author Response

(The authors gave the same response as above.)

Round 2

Reviewer 1 Report

In the revised version of the manuscript, Yu et al. provided answers to the previous reviewers’ concerns. I still have some concerns.

The authors need to clearly mention how many spectra were used for the training of the CNN; the evaluation values of the training process need to be shown.

I suggest tuning down the manuscript. As the authors mentioned, this work is an incentive study an indication of the feasibility of machine learning. Words like “100% reproducibility in double blind experiments” are biased and annoying because the sample size is so small.

Author Response

We thank the reviewer for their suggestion. We have modified the CNN section as follows - 

We started with a data set size of over 1200 spectra, i.e. at-least 80 spectra per sample. To increase the training set, some data augmentation methods on the training data is done: (i) Random shifting of each spectrum by a few (1~2) wavenumbers. (ii) Introduction of a random noise onto each spectrum. (iii) We also randomly produced linear combinations of spectra collected from the same mapping procedure.

Furthermore, we have also toned down the statements and clearly mentioned and acknowledged the small sample size issue in our conclusion and main body.

Reviewer 3 Report

Dear Authors.

The revision adequately address the concerns expressed in last review.

So, I recommend that this revised manuscript can now be recommended for publication (Accept as is).

Author Response

We would like to thank the reviewer for their recommendation to publish.